# TGF-β-Upregulated *Lnc-Nr6a1* Acts as a Reservoir of miR-181 and Mediates Assembly of a Glycolytic Complex

**DOI:** 10.3390/ncrna8050062

**Published:** 2022-09-15

**Authors:** Salvador Polo-Generelo, Belén Torres, José A. Guerrero-Martínez, Emilio Camafeita, Jesús Vázquez, José C. Reyes, José A. Pintor-Toro

**Affiliations:** 1Department of Cell Signalling, Centro Andaluz de Biología Molecular y Medicina Regenerativa (CABIMER-CSIC), 41092 Sevilla, Spain; 2Cardiovascular Proteomics, Centro Nacional de Investigaciones Cardiovasculares (CNIC), 28029 Madrid, Spain; 3CIBER de Enfermedades Cardiovasculares (CIBERCV), 28029 Madrid, Spain

**Keywords:** lncRNA, miRNA-host genes, microprocessor, cell adhesion, anoikis, glycolytic scaffold

## Abstract

Long non-coding RNAs (lncRNAs) have emerged as key regulators in a wide range of biological processes. Here, we identified a mouse miRNA-host gene lncRNA (*lnc-Nr6a1*) upregulated early during epithelial-to-mesenchymal transition (EMT). We show that when lncRNA is processed, it gives rise to two abundant polyadenylated isoforms, *lnc-Nr6a1-1* and *lnc-Nr6a1-2*, and a longer non-polyadenylated microprocessor-driven lnc-pri-miRNA containing clustered pre-miR-181a2 and pre-miR-181b2 hairpins. Ectopic expression of the *lnc-Nr6a1-1* or *lnc-Nr6a1-2* isoform enhanced cell migration and the invasive capacity of the cells, whereas the expression of the isoforms and miR-181a2 and miR-181b2 conferred anoikis resistance. *Lnc-Nr6a1* gene deletion resulted in cells with lower adhesion capacity and reduced glycolytic metabolism, which are restored by *lnc-Nr6a1-1* isoform expression. We performed identification of direct RNA interacting proteins (iDRIP) to identify proteins interacting directly with the *lnc-Nr6a1-1* isoform. We defined a network of interacting proteins, including glycolytic enzymes, desmosome proteins and chaperone proteins; and we demonstrated that the *lnc-Nr6a1-1* isoform directly binds and acts as a scaffold molecule for the assembly of ENO1, ALDOA, GAPDH, and PKM glycolytic enzymes, along with LDHA, supporting substrate channeling for efficient glycolysis. Our results unveil a role of *Lnc-Nr6a1* as a multifunctional lncRNA acting as a backbone for multiprotein complex formation and primary microRNAs.

## 1. Introduction

EMT is a reversible transdifferentiation process in which epithelial cells lose their characteristics and instead acquire attributes of mesenchymal cells. During EMT, cell-cell junctions are disrupted, the actin cytoskeleton is extensively reorganized and the cells acquire increased migratory and invasive characteristics [1,2]. EMT is essential in several physiological and pathological processes, including embryonic development, wound healing, organ fibrosis and cancer progression [3]. The EMT process entails substantial changes in gene expression. Beyond the well-stablished transcriptional reprogramming during EMT, posttranscriptional mechanisms, such as regulation by alternative pre-mRNA splicing and non-coding RNA, have an important role and provide additional layers of complexity in gene regulation [4,5,6].

Long non-coding RNAs (lncRNAs) are defined as RNA transcripts longer than 200 nucleotides, with no apparent protein-coding potential. More than 16,000 lncRNA genes have been identified in the human genome, on the basis of GENCODE annotations (Human GENCODE Release, v27), but only a small percentage of these have been functionally characterized. Notably, their expression is strikingly cell-type or tissue-restricted [7]. Studies on lncRNAs have demonstrated that they serve important roles in gene regulation as molecular scaffold or guides modulating nucleic acid-nucleic acid, nucleic acid-protein or protein-protein interactions, or competing endogenous RNAs that bind proteins or microRNAs (miRNAs); and as enhancers of gene expression when transcribed within enhancer regions or their neighboring loci [8,9]. Some lncRNAs harbor miRNAs within their exonic or intronic sequences, and hence are processed to produce miRNAs that promote mRNA decay or repress target mRNAs. For example, lncRNA-H19 exon1 encodes miR-675 miRNA that suppresses the translation of the *Igf1r* gene, inhibiting cell proliferation [10]; miR-100 and miR-125b generated from the lncRNA MiR100HG mediate cetuximab resistance via Wnt/b-catenin signaling [11]; PVT1 lncRNA, which is processed from the MIRHG harboring the miR-1204, miR-1205, miR-1206, miR-1207-5p, miR-1207-3p and miR-1208 cluster positively regulates *c-Myc* expression and activity [12]. A major question that needs to be answered is whether these lncRNAs are merely serving as primary miRNA units for producing miRNAs, or if they might also generate mature lncRNAs that have independent roles. 

In the present work, we studied a TGF-β-responsive lncRNA, we named *Lnc-Nr6a1*, expressed early during EMT. Two polyadenylated isoforms, *lnc-Nr6a1-1* and *lnc-Nr6a1-2*, and a longer non-polyadenylated isoform were generated from *Lnc-Nr6a1* genomic loci. The non-polyadenylated isoform contains pri-miR181a2 and pri-miR181b2 sequences, which are Dicer processed generating mature miR-181a2 and miR-181b2. We showed that migration and invasiveness of the cell increase by ectopic expression of the polyadenylated isoforms, whereas anoikis resistance increases by expression of the non-polyadenylated isoform, proving that transcripts generated from *Lnc-Nr6a1* genomic loci perform independent functions. *Lnc-Nr6a1* gene depletion causes defects in cell adhesion and metabolic changes that are reverted by exogenous expression of the polyadenylated *lnc-Nr6a1-1* isoform. We demonstrated that this lncRNA isoform interacts directly with the proteins of several functional groups and serves as molecular scaffold enabling the formation of a multienzyme complex between ENO1, ALDOA, GAPDH, and PKM glycolytic enzymes, along with LDHA, and contributing to a higher level of glycolysis.

## 2. Results

### 2.1. Upregulation of Lnc-Nr6a1 and Embedded miR-181a2 and miR-181b2 during EMT

To systematically identify lncRNAs involved early in the EMT process, transcriptional profiling was performed in untreated NMuMG mammary epithelial cells and NMuMG cells treated with TGF-β for 2 h using microarrays that covered 31,423 annotated lncRNAs and also 25,376 protein-coding transcripts [5]. Among PCR-confirmed candidates, *Nr6a1os* was one of the most significantly induced non-coding RNAs. This gene maps on chromosome 2 and is transcribed from the third intron of the *Nr6a1* gene in the opposite sense. Two isoforms of 650 bp and 645 bp, here named *lnc-Nr6a1-1* and *lnc-Nr6a1-2*, respectively, were induced early after TGF-β treatment. The isoform *lnc-Nr6a1-1* is composed of three exons separated by introns of 16,600 and 13,806 bp, and *lnc-Nr6a1-2* of two exons and an intron of 14,699 bp. MiR-181a2 and miR-181b2 genes are located at 314 and 1410 bp downstream, respectively, of the 3′ end of *lnc-Nr6a1-1* (Figure 1A). Exon2- and miR-181a2/b2-specific probes showed that an unprocessed lnc-pri-miR-181a2/b2 transcript was generated containing spliced exon1, exon2 and exon3 and an unprocessed fragment with embedded miR-181a2 and miR-181b2 genes (Figure 1A–C). Quantitative reverse-transcription PCR (RT-qPCR) of poly(A**^+^**)- and poly(A**^−^**)-selected RNA showed that the *lnc-Nr6a1-1* isoform was found in both RNA populations, in contrast to the *lnc-Nr6a1-2* isoform that was found only in the poly(A**^+^**) RNA population. We validated the enrichment using primers specific for a polyadenylated mRNA (*Hprt)* and non-polyadenylated RNA (*U6*) (Figure 1D). To map lnc-pri-miR-181a2/b2 3′ ends at nucleotide resolution, we developed both a poly(A**^+^**) tail–dependent and a poly(A**^+^**) tail–independent 3′ rapid amplification of cDNA ends (3′ RACE) technique. Two poly(A**^+^**)-dependent polyadenylation sites were mapped, one of them located after the second exon of the *lnc-Nr6a1-2* isoform, according to the annotation ENSMUST00000129089.2 from GENECODE, with a canonical AATAA polyadenylation signal occurring 35 nt upstream from the poly(A) tail, and the other one located after the third exon of *lnc-Nr6a1-1*, 49 nucleotides longer than the ENSMUST00000151229.2 transcript annotation, with a noncanonical AATTAAA polyadenylation signal occurring 19 nt upstream from the poly(A) tail (Appendix A). Four poly(A**^+^**) tail-independent polyadenylation sites were mapped, corresponding to the sites of Drosha cleavage on the 5′ arm of the pre-miR-181a2 and pre-miR-181b2, respectively; and the site of Drosha cleavage on the 3′ arm of the pre-miR-181a2 and pre-miR-181b2, respectively (Appendix A). These results indicate that the *lnc-Nr6a1-1* isoform is transcribed as a polyadenylated RNA and also as a precursor non-polyadenylated RNA, which would be processed by the Microprocessor complex providing pre-miR-181a2 and pre-miR-181b2 molecules and a lncRNA molecule that would be degraded since it is not detected using specific primers. We determined by 5′ rapid amplification of cDNA ends (5′ RACE) that *lnc-Nr6a1-1* is a transcript 75 nucleotides longer at the 5′ end than the ENSMUST00000151229.2 transcript annotation (Appendix A and Figure 2C,D). *Lnc-Nr6a1* is located at the third intron of the *Nr6a1* gene, and its transcription could play a *cis*-regulatory role in the expression of this gene. However, *Nr6a1* expression, unlike *lnc-Nr6a1* isoforms and miR-181a2/b2, is hardly affected by treatment with TGF-β (Figure 1E–G). Likewise, *Nr6a1* transcript levels were low in most tissues and organs analyzed except in testis, while expression profiles of *lnc-Nr6a1* isoforms in multiple adult tissues showed that they were expressed notably in skeletal muscle, and differed markedly in other tissues or organs; therefore, *lnc-Nr6a1-1* was expressed in the eyes while *lnc-Nr6a1-2* was expressed in bone marrow and the thymus, suggesting that they could regulate different biological processes (Figure 1H,I). Together these data demonstrate that the expression of *Lnc-Nr6a1* and *Nr6a1* genes are not matched. Sequence analysis using ORF Finder (National Centre for Biotechnology Information) predicted open reading frames smaller than 43 amino acids without a valid Kozack sequence. In addition, a low-coding potential score (−0.83 and −0.86) and a low-coding probability (0.028 and 0.028) returned by the Coding Potential Calculator software [13] and Coding-Potential Assessment Tool [14], respectively, confirmed a negligible protein coding potential for *lnc-Nr6a1-1* and *lnc-Nr6a1-2*, respectively (Appendix A).

A human *Lnc-NR6A1* orthologue transcript produced by the syntenic locus in chromosome 9 displays an overall 60% and 68% sequence identity with *lnc-Nr6a1-1* and *lnc-Nr6a1-2*, respectively. It is composed of 2 exons of 330 and 387 bp and an intron of 40,220 bp that harbors mir-181a2 and mir-181b2 genes. In a similar way to the murine counterpart, expression of the human transcript was induced in MCF-10A human breast epithelial cells and primary human omental mesothelial cells after TGF-β treatment (Appendix A).

Signaling pathways involved in *lnc-Nr6a1* isoforms induction by TGF-β were investigated. Smad4 is a central mediator of the Smad-dependent canonical pathway. Two different Dicer-substrate short interfering RNAs were used for silencing Smad4 and the knockdown efficiencies were tested by Western blotting. Upregulation of *lnc-Nr6a1* isoforms by TGF-β was not affected in Smad4-suppressed cells, indicating TGF-β-Smad signaling independency of TGF-β response (Figure 2A). TGF-β non-canonical pathways, extracellular signal-regulated kinases (ERKs), p38 mitogen-activated protein kinase (p38 MAPK) and protein kinase B (AKT) are activated very rapidly in TGF-β-treated NMuMG cells [15]. Blocking of p38 MAP kinase and AKT signaling pathways had no effect on TGF-β-upregulation of *lnc-Nr6a1-1* (Figure 2B,D), whereas blocking the MEK-ERK1/2 signaling pathway inhibited its expression (Figure 2C). Remarkably, *lnc-Nr6a1-2* expression was specifically abolished by GSK6906693, an inhibitor of AKT (Figure 2D). Taken together these results indicate that each isoform is regulated by a different signaling pathway. The early upregulation of *lnc-Nr6a1-1/2* isoforms suggested an active role in the start of the EMT program. To determine whether they were direct targets of ERK and AKT signaling pathways, expression levels of *lnc-Nr6a1-1/2* isoforms were analyzed in cells treated with TGF-β in the presence or absence of cycloheximide, an inhibitor of protein synthesis. RT-qPCR results showed upregulation of both isoforms in the presence of cycloheximide, whereas upregulation of an indirect target control, SIP1, was blocked (Figure 2E). These data suggest that *lnc-Nr6a1-1* and *lnc-Nr6a1-2* isoforms are direct targets of ERK1/2 and AKT pathways, respectively.

### 2.2. Lnc-Nr6a1 Depletion Reduces Cell Adhesion and Alters Gene Expression

To examine the physiological relevance of *Lnc-Nr6a1* expression in EMT, we used CRISPR-Cas9 technology to remove *Lnc-Nr6a1* transcription. NMuMG cells were edited by CRISPR using two pairs of single guide RNAs (sgRNAs) located in the promoter and first exon of *Lnc-Nr6a1,* respectively, obtaining a deletion of 600 bp (Figure 3A). Complete inhibition of *lnc-Nr6a1* isoforms expression was achieved in homozygous clones (Figure 3B,C). Furthermore, disruption of the *Lnc-Nr6a1* host gene expression affected the mi181a2/b2 expression (Figure 3D). It should be noted that expression of identical mature miR-181a and miR-181b, encoded by mir-181a1 and mir-181b1 genes on chromosome 1, was not affected by TGF-β treatment. Previous chromatin immunoprecipitation followed by deep sequencing (ChIP-seq) assays in the same cell line [16] showed a remarkable H3K4me3 and H3K27ac-enriched region with high chromatin accessibility (via ATAC-seq) around the transcription start site of the *Lnc-Nr6a1* gene. Neither H3K4me3 and H3K27ac enrichment nor chromatin accessibility were found in the miR181a2/b2 region (Appendix A). In addition, querying the PanCancer Atlas expression data reveals a strong co-expression relationship between *Lnc-Nr6a1* and embedded miR-181a2/b2 in different tumors (Appendix A). Altogether, these results support that up-regulation of miR-181a2/b2 and *Lnc-Nr6a1* after TGF-β treatment are driven by activation of the same promoter. To reveal the functional importance of the *lnc-Nr6a1* transcripts, we performed transcription profiling of the *Lnc-Nr6a1-*depleted cells. *Lnc-Nr6a1* knockout resulted in statistically significant changes in the expression levels of 218 genes (adjusted *p*-value < 0.05 and |log_2_FC| > 0.5) compared with the control cells; among them 95 and 123 were downregulated and upregulated, respectively (Appendix A and Figure 3E). Some of these changes were confirmed by RT-qPCR, as downregulation (>100-fold) of *Dsg2*, *Fam111a*, *Mal2*, *FoxF1* and upregulation of *Prune*, *Pkp1*, *Adgrl3*, (Appendix A). Gene ontology analysis of the downregulated genes in *Lnc-Nr6a1^−/−^* cells showed an enrichment in genes involved in cell adhesion and the glycolytic process (Figure 3F and Appendix A). Next, we investigated the effect of *Lnc-Nr6a1* knock-down on the adhesion capacity of NMuMG cells to substrate and cells. *Lnc-Nr6a1*-depleted cells showed a lower capacity to adhere to the substrate; this effect was observed over a long and short time (Figure 4A,B). Furthermore, neither cell cycle changes nor alterations in proliferation were observed in *Lnc-Nr6a1*-depleted cells (Appendix A). In order to determine the effect of *Lnc-Nr6a1* absence on cell-cell adhesion, monolayers of NMuMG cells were incubated with calcein-labelled *Lnc-Nr6a1*-depleted cells or control cells for 2 h. Flow cytometry analysis showed a lower ability of *Lnc-Nr6a1*-depleted cells to adhere to cells (Figure 4D,E). The loss of adhesion capacity to substrate and cells were partially recovered after expressing *lnc-Nr6a1-1* isoform (Figure 4C–E). Together, the combination of lncRNA profiling and functional assays identified *Lnc-Nr6a1* as a lncRNA involved in cell adhesion to substrate and cells.

Since *Lnc-Nr6a1* is induced early by TGF-β, we studied whether expression changes produced by TGF-β were dependent on the presence of *Lnc-Nr6a1*. We analyzed the transcriptomic profiles of the control cells versus *Lnc-Nr6a1*-depleted cells after 2 h of TGF-β treatment (Appendix A). In total, 895 genes were upregulated and 747 genes downregulated in the control cells and 641 and 459 genes, respectively, in the *Lnc-Nr6a1*-depleted cells (adjusted *p*-value < 0.05 and |log_2_FC| > 0.5). We found 44% and 27% of upregulated and downregulated genes, respectively, by TGF-β were common in both types of cells; therefore, their regulation was independent of the presence or absence of *Lnc-Nr6a1*. These genes were enriched in those involved in EMT, phosphatidylcholine biosynthetic process, DNA replication checkpoint and chromatin silencing (Figure 5A,B). This result indicates that 56% and 73% of genes induced or inhibited, respectively, by TGF-β are dependent on the presence of *lnc-Nr6a1* and slightly enriched in genes involved in histone modification and ribosomal RNA maturation. Likewise, *Lnc-Nr6a1* depletion caused induction or inhibition of new genes by TGF-β (39% and 55%, respectively) enriched mainly in meiotic chromosome condensation, DNA replication initiation and negative regulation of chromatin silencing. Together, these data indicate that *Lnc-Nr6a1* depletion affects differential gene expression mediated by TGF-β; however, expression of genes involved in EMT was independent on the presence of *Lnc-Nr6a1*. Forced expression of *lnc-Nr6a1* isoforms neither revealed apparent morphological changes in cells nor significant changes in expression levels of the EMT markers. Furthermore, neither cell cycle nor proliferation were affected in isoform-overexpressing cells (Appendix A). However, wound-healing assays revealed that cells overexpressing either of the two isoforms migrated much faster than control cells, but the effects were not additive. Notably, cell migration was not affected by *Lnc-Nr6a1* depletion (Figure 5C–E, Appendix A).

Since one of the *lnc-Nr6a1* isoforms is transcribed as a precursor containing miR-181a2 and miR-181b2 genes, we investigated the effect of precursor over-expression using a CRISPR/Cas9 synergistic activation mediator (SAM) system (Figure 6A) [17]. To this end, we designed 10 single-guide RNA (sgRNAs) targeting regions close to the transcription start site (Appendix A) and found that sgRNA 2 and sgRNA 8 notably increased both *lnc-Nr6a1* isoforms and miR-181 genes expression, as shown by qPCR analysis (Figure 6B), supporting that regulation of *Lnc-Nr6a1* and miR-181a2/b2 is driven by the same promoter. Cells overexpressing the *Lnc-Nr6a1* gene showed the epithelial phenotype. However, both immunofluorescence and immunoblotting analysis revealed an increased expression of the mesenchymal marker N-cadherin and a decrease in the epithelial marker E-cadherin expression (Figure 6C,D). As *Lnc-Nr6a1* upregulation induced expression changes associated with mesenchymal cells, we wondered whether *Lnc-Nr6a1* upregulation could affect features of the EMT program. Neither proliferation nor adhesion capacity were affected in precursor-overexpressing cells (Appendix A); however, wound healing and Matrigel invasion assays revealed that these cells migrated significantly faster and were more invasive than *lnc-Nr6a1-1* overexpressing cells (Figure 6E–G, Appendix A). A feature of EMT programs is they contribute to the resistance of cells to anoikis. We determined the percentage of apoptotic cells in *lnc-Nr6a1-1*- and *Lnc-Nr6a1*-upregulated cells growing in the presence of methylcellulose and on Poly-HEMA-coated plates to prevent both adherence among them and their adherence to plastic, thereby eliciting anoikis. Cells overexpressing *Lnc-Nr6a1* presented higher anoikis resistance than cells overexpressing the *lnc-Nr6a1-1* isoform and control cells (Figure 6H). These results indicate, on the one hand, that miR-181a2/b2 expression increases the migratory and invasive ability of the cells and confers resistance to anoikis; and on the other hand, that the mature transcripts and embedded miRNAs from *Lnc-Nr6a1* genomic loci perform independent functions.

### 2.3. Lnc-Nr6a1 Protein Interactome

To explore the molecular mechanisms underlying the biological activity of *Lnc-Nr6a1*, we searched for proteins that were associated with *lnc-Nr6a1* using the iDRIP in vivo technique (identification of direct RNA interacting proteins) [18] (Figure 7A). After UV crosslinking of NMuMG cells, *lnc-Nr6a1*-protein complexes were purified using *lnc-Nr6a1* antisense biotinylated oligonucleotides. Different pools of biotinylated antisense probes were tested against the *lnc-Nr6a1* isoforms. We achieved a pool of oligonucleotides that specifically retrieved over 30% of the *lnc-Nr6a1-1* complexes, without enrichment of *lnc-Nr6a1-2* complexes or housekeeping *Hprt* mRNA. A *LacZ* oligonucleotide pool that targeted the *lacZ* mRNA did not retrieve *Hprt* nor *lnc-Nr6a1-1* RNAs (Figure 7B). It should be mentioned that we have been struggling to achieve a pool of biotinylated oligonucleotides that specifically retrieves the *lnc-Nr6a1-2* isoform, but all attempts to solve this problem have so far been unsuccessful. After washing the complexes under denaturing conditions and treating them with DNAse I, the eluted proteins were analyzed by mass spectrometry. A total of 42 proteins were found to be enriched (≥2-fold) over background (Figure 7C and Appendix A). These proteins belonged to different functional groups: protein-folding chaperones (HSP90AA1, HSPA5, HSPA1B, HSPA8, 14-3-3s/g/e/z/d), lamin binding (HNRNPK, PKP1), cell-cell junction organization (DSP, PKP1, ACTN4, JUP), translational elongation (EEF1A1, EEF2, EEF1G), glycolytic process (ALDOA, ENO1, PKM, GAPDH) and intermediate filament-based process (DSP, PLEC, PKP1, EVPL). Interactions of the most enriched proteins were confirmed by RNA immunoprecipitation (RIP) assay using specific and non-specific IgG antibodies (Figure 7D). Prelamin A/C was the most enriched protein by *lnc-Nr6a1-1* probes. The subcellular location of *lnc-Nr6a1* isoforms using *Hprt*, *U3* and *Xist* RNAs as indicators for cytosolic and soluble and insoluble nuclear fractions, respectively, resulted in nuclear enrichment located mainly in the chromatin fraction (Figure 7E). Since a significant proportion of *lnc-Nr6a1-1* was located at the nuclear insoluble fraction and one of the major components of this fraction is chromatin, we studied whether *lnc-Nr6a1-1* isoform could be interacting with DNA and modulating gene expression. The genome-wide binding profile of *lnc-Nr6a1-1* was mapped using the ChIRP technique (chromatin isolation via RNA immunoprecipitation) [19]. This technique allows the discovery of RNA-associated DNA sequences and mapping of genomic binding sites of chromatin-associated lncRNAs with high sensitivity and low background. Target lncRNA from glutaraldehyde-crosslinked cells was affinity captured using antisense-biotinylated oligonucleotides, and the lncRNA-associated DNA chromatin used to create a sequencing library. Under the conditions used for ChIRP, more than 30% of the *lnc-Nr6a1-1* RNA was retrieved from the control cells. Results of ChIRP-seq analysis, using *Lnc-Nr6a1*-depleted cells as the negative control, showed that there were no specific DNA binding sites for the *lnc-Nr6a1**-1* isoform. Thus, we concluded that nuclear *lnc-Nr6a1**-1* molecules bind Lamin A/C in regions not associated with the genome. Lamins associate with chromatin through large domains termed lamina-associated domains or LADs, which contain mostly inactive genes and are located at the nuclear periphery [20]. Since *lnc-Nr6a1**-1* isoform interacted with Lamin A/C, but not with chromatin, we wondered if the depletion of *Lnc-Nr6a1* could affect the specific Lamin A/C-associated genomic domains (LADs). Genomic regions that interact with Lamin A/C in the control and *Lnc-Nr6a1*-depleted cells were mapped by chromatin immunoprecipitation coupled with deep sequencing (ChIP-seq). A total of 619 LADs were mapped in the control cells, 148 of them were partially lost in *Lnc-Nr6a1*-depleted cells, and 45 new LADS appeared in the *Lnc-Nr6a1*-depleted cells (Appendix A). Genes associated with these LADs were identified (Appendix A) and gene ontology analysis of genes associated to partially lost LADs showed enrichment in genes involved in smell sensory perception and chemotaxis (Appendix A). Moreover, comparison between LADs-associated genes and expression changes detected in *Lnc-Nr6a1*-depleted cells showed a match in only five genes (*p*-value = 0.17) (Appendix A), inferring that the absence of interaction of *lnc-Nr6a1-1* with lamin A/C is not the direct cause of the gene expression changes in depleted cells.

Our finding that glycolytic proteins ALDOA, ENO1, PKM, GAPDH, along with LDHA, interact with *lnc-Nr6a1-1* is outstanding. Covalent bonds between *lnc-Nr6a1**-1* and interacting proteins are direct because they were created by UV crosslinking. This result strongly suggested that *lnc-Nr6a1**-1* could act as a scaffold molecule to gather glycolytic proteins to promote glycolysis by substrate channeling. To test this hypothesis, we measured the glycolysis metabolic process in the control and *Lnc-Nr6a1*-depleted cells using the Seahorse Analyzer, which provides accurate measurements of glycolytic rates for basal conditions and compensatory glycolysis following mitochondrial inhibition. We found that the knockout of *Lnc-Nr6a1* significantly reduced basal glycolysis and compensatory glycolysis and that these effects were all reverted by overexpression of the *lnc-Nr6a1-1* isoform (Figure 8A,B).

Next, we analyzed whether the interaction of *lnc-Nr6a1-1* with each of the proteins occurred independently or whether the *lnc-Nr6a1-1* isoform forms a glycolytic complex with these enzymes. To this end, coimmunoprecipitation experiments were carried out on UV-crosslinked extracts from control cells and *Lnc-Nr6a1*-depleted cells using an antibody against ENO-1 and the immunoprecipitated proteins were subjected to quantitative mass spectrometry. The analysis results showed the enrichment of glycolytic enzymes ALDOA, PKM, and GAPDH, along with LDHA previously detected by theiDRIP technique, in addition to the enrichment of the glycolytic enzymes TPI1, PGAM1 and PGK1 (Appendix A). Furthermore, quantification of immunoprecipitated proteins revealed a markedly lower amount of immunoprecipitated glycolytic enzymes in *Lnc-Nr6a1*-depleted cells (Figure 8C). These data indicate that *lnc-Nr6a1-1* may function as a molecular scaffold for glycolytic multi-enzymatic complex formation (Figure 8D) and this promotes enhanced catalytic efficiency.

## 3. Discussion

In this study, we analyzed a lncRNA, *Lnc-Nr6a1*, which is induced early and abundantly during EMT. Two main isoforms are generated in response to TGF-β, with very similar expression levels. The expression of each one is regulated by different pathways: the *lnc-Nr6a1-1* isoform by MEK-ERK1/2 and *lnc-Nr6a1-2* isoform by AKT, routes that are simultaneously activated by TGF-β, and that would indicate that the isoforms could be involved in specific processes that respond to each of these routes. Each isoform has a polyadenylated transcript, but in the case of the *lnc-Nr6a1-1* isoform it is also present as a non-polyadenylated precursor containing processed exons, miR-181a2 and miR-181b2 genes; as well as genomic sequences between the 3′-end of polyadenylated *lnc-Nr6a1-1* and 5′-end of the miR-181a2 gene and the genomic sequence between both miRNAs. The transcriptional termination of the non-polyadenylated isoform and processing of pre-miR181a2 and pre-miR-181b2 would be mediated by the microprocessor complex [21,22]. Studies reported that the cleavage of the precursor transcript by Drosha occurs cotranscriptionally and sometimes even before the splicing of introns [23,24]; however, our results indicate that the cleavage of the non-polyadenylated *lnc-Nr6a1-1* transcript by Drosha occurs after splicing of the introns. Regarding the resulting lncRNA molecule after processing by Drosha of the pre-miRNAs, the analysis of the UCSC genome browser shows the absence of ESTs between the 3′-end of polyadenylated *lnc-Nr6a1-1* and 5′ end of miRNA loci suggesting that *lnc-Nr6a1* non-polyadenylated mature transcript resulting from processing by Drosha is degraded [21]; therefore, *lnc-Nr6a1-1* and *lnc-Nr6a1-2* isoforms and miR-181a2 and miR-181b2 would be the molecules generated from the *Lnc-Nr6a1* gene. The presence of polyadenylated and non-polyadenylated forms would make it possible to regulate the expression of these molecules, allowing high levels of one of them to be achieved without concomitant generation of high levels of another unwanted one. As has been described for other miRNA-host lncRNAs [11,22,25], *Lnc-Nr6a1* carries out specific functions that are independent of its role as a miRNA precursor, thus the loss of adhesion capacity shown by *Lnc-Nr6a1*-depleted cells is rescued by exogenous expression of the *lnc-Nr6a1-1* isoform; in the same way, the glycolytic activity in *Lnc-Nr6a1*-depleted cells reaches levels similar to those of the control cells when overexpressing the *lnc-Nr6a1-1* isoform; and also, overexpression of any of the polyadenylated isoforms increases cell mobility and invasiveness. However, anoikis resistance only increases when miR-181a2 and miR-181b2 are expressed. Therefore, the *Lnc-Nr6a1* gene, in addition to providing miR-181a2/b2 molecules, generates polyadenylated lncRNA molecules with independent functions.

The in vivo study of proteins interacting with the *lnc-Nr6a1-1* isoform reveals that this isoform interacts with different proteins, some of them participating in the same process or mechanism, suggesting that *lnc-Nr6a1-1* could act as a scaffold molecule by recruiting or titrating the interacting proteins. Lamin A/C is the most enriched *lnc-Nr6a1-1*-interacting protein. Lamin A/C is a structural component of the nuclear lamina that regulates genome organization and gene expression. Lamin A/C associates with chromatin through large domains termed lamina-associated domains or LADs, which are located at the nuclear periphery [26]; however, *lnc-Nr6a1-1* does not interact with DNA. Therefore, its interaction with Lamin A/C would take place in regions where Lamin A/C does not interact with DNA, such as the nucleoplasm, where significant levels of A-type lamins have been found [27]. The LADs present in the cell change as gene expression changes; however, in *Lnc-Nr6a1*-depleted cells only partial changes in LADs are detected, which do not correspond to gene expression changes detected in the *Lnc-Nr6a1*-depleted cells. This would indicate that *lnc-Nr6a1-1* carries out functions in the nucleus, unknown at the moment, but it would not act as a transcriptional regulator. *Lnc-Nr6a1-1* interacts directly with the proteins plakophilin, plakoglobin and desmoplakin; the three main components of the desmosome [28]. Desmoplakin couples intermediate filaments to the desmosome and its amino-terminus binds directly to plakoglobin and the plakophilin [29]. The direct interaction of *lnc-Nr6a1-1* with these proteins marks it as an additional component of the desmosome that could facilitate the formation of the complex. This could explain the cell-cell adhesion decrease in *Lnc-Nr6a1*-depleted cells and subsequent recovery after exogenous expression of the *lnc-Nr6a1-1* isoform. It is increasingly evident that glycolysis proteins associate to form multienzyme complexes, especially under stress conditions to meet excessive energy demand. Studies on yeast have revealed that glycolysis enzymes colocalize into non-membrane-bound granules, named glycolytic bodies or G bodies, in response to hypoxia [30]; also, in neurons of *Caenorhabditis elegans*, analogous granules are formed to sustain changes in the activity of synapses [31]. These structures are formed by phase separation and RNA molecules appear to be responsible for concentrating and sequestering protein components through multiple binding sites [32,33]. A large number of studies have shown that lncRNAs can affect genes involved in glucose metabolism by directly regulating the glycolytic enzymes and glucose transporters or indirectly modulating the signaling pathways [34]. We have found that ENO1, ALDOA, GAPDH, and PKM glycolytic enzymes, along with LDHA, interact in vivo directly with *lnc-Nr6a1-1* isoform in non-transformed NMuMG cells in normal conditions. Glycolytic metabolism is affected in *Lnc-Nr6a1*-depleted cells; thus, these cells showed lower basal and reserve glycolytic activity than control cells; but the glycolytic activity was recovered upon exogenous expression of the *lnc-Nr6a1-1* isoform. We propose that *lnc-Nr6a1-1* act as a scaffold molecule interacting directly with these enzymes. We must underline the fact that these enzymes were coprecipitated with antibody against ENO1 irrespective of the presence of *lnc-Nr6a1-1*; however, in *Lnc-Nr6a1*-depleted cells, coprecipitated enzymes were markedly reduced, indicating that *lnc-Nr6a1-1* supports formation of a multi-protein complex of glycolytic enzymes and that this results in enhanced catalytic efficiency. Coimunoprecipitation of this pool of enzymes from UV-crosslinked *Lnc-Nr6a1-*depleted cells indicates that other RNAs could jointly contribute to the glycolytic complex formation. In fact, gLINC lnc-RNA was recently reported to act as a backbone for complex formation between PGK1/ENO1/ and PKM2, along with LDHA, promoting glycolytic flux and ATP production [35]; in the same way *NEAT1* lncRNA forms a scaffold bridge for the assembly of the glycolytic complex PGK1, PGAM and ENO1 promoting a glycolytic state in breast cancer that is associated with more invasive and higher-grade tumors [36]. The contribution of other lncRNAs in the formation of the glycolytic complex would support that the enzymes TPI1, PGK1 and PGAM1 are co-immunoprecipitated by ENO1, despite not being present in the *lnc-Nr6a1-1* interactome. A hallmark of cancer cells is metabolic reprogramming to generate energy through increased glycolysis even in the presence of oxygen, known as the Warburg effect [37,38]. High levels of *Lnc-Nr6a1* are observed in most of the tumors analyzed (Appendix A), which could be associated with the formation of multienzyme complexes and increased glycolytic activity in cancer cells. Our results show different functions for *Lnc-Nr6a1*. On the one hand, by means of its non-polyadenylated transcript, *Lnc-Nr6a1* serves as a miRNA reservoir, giving rise to miRNAs miR-181a2 and miR-181b2; on the other hand, polyadenylated *lnc-Nr6a1-1* transcript is involved in cell adhesion and glycolytic metabolism. Additional non-overlapping mechanisms for the *lnc-Nr6a1-2* isoform are not ruled out. Further explorations of the isoform-specific interactome will unveil the full biological activities of this lncRNA.

## 4. Materials and Methods

### 4.1. Reagents

The following antibodies were used in our study: anti-ERK1/2 (Cat# 61-7400; Thermo Fisher Scientific, Waltham, MA, USA), anti-phospho-ERK1/2 (9101; Cell Signaling Technology, Danvers, MA, USA), anti-phospho-p70 S6 kinase (Cat# 9206; Cell Signaling Technology), anti-phospho (Ser/Thr) AKT substrate (Cat# 9611; Cell Signaling Technology), anti-phospho-p85 (Cat# 9206; Cell Signaling Technology), anti-Smad4 (Cat# sc-7966; Santa Cruz Biotechnology, Santa Cruz, CA, USA), anti-E-cadherin (Cat# 610181; BD Transduction Laboratories, BD Biosciences, San Jose, CA, USA), anti-N-cadherin (Cat# 610921; BD Transduction Laboratories), anti-Vimentin (Cat# sc-32322; Santa Cruz), anti-Snail1 (Cat# C15D3; Cell Signaling Technology), anti-Enolase1 (Cat# 3810; Cell Signaling Technology), anti-GAPDH (Cat# sc-47724; Santa Cruz), anti-ALDOA (Cat# 11217-1-AP; Proteintech, Proteintech Europe Ltd, Manchester, UK), anti-Pyruvate kinase (Cat# 3106; Cell Signaling Technology), anti-Lamin A/C (Cat# sc-376248; Santa Cruz), anti-g-Catenin (Cat# A0963; ABclonal, Woburn, MA, USA), anti-Plakophilin (Cat# sc-33636; Santa Cruz), anti-β-Actin (Cat# A1978; Sigma-Aldrich, St Louis, MO, USA). GSK690693, U0126 and SB-203580 were purchased from Selleck Chemicals (Houston, TX, USA). TGF-β was purchased from PeproTech (London, UK).

### 4.2. Cell Culture

NMuMG cells were grown in Dulbecco’s modified Eagle’s medium containing 10% fetal bovine serum, 2 mM L-glutamine, antibiotics (50 U/mL penicillin, 50 lg/mL streptomycin) and insulin (10 µg/mL) at 37 °C in a 5% CO_2_ humidified, 95% air incubator.

### 4.3. Lentiviral Production and Infection Assay

Lentiviral production and infection assays were performed as described previously [5]. For ectopic expression of *lnc-Nr6a1-1* or *lnc-Nr6a1-2*, NMuMG cells were infected with pHRSIN-DUAL-*lnc-Nr6a1-1* or *lnc-Nr6a1-2* lentiviral supernatants (MOI 2), respectively, containing 4 µg/mL polybrene. The dual-promoter lentivector pHRSIN-DUAL (also known as pHRSIN-CSGWdINotI_pUb_Em) was kindly provided by Mary K. Collins (Okinawa Institute of Science and Technology, Okinawa, Japan).

### 4.4. Western Blot Analysis

Protein was extracted with a RIPA buffer and western blotting was performed according to the standard protocols. Signals were detected with an ECL kit (GE Healthcare, Buckinghamshire, UK) and visualized and quantified using the ChemiDoc MP Imaging system (Bio-Rad, Hercules, CA, USA).

### 4.5. Proliferation, Migration and Invasion Assays

For growth curves, cells were harvested and transferred into 16-well E-plates, which contained electrodes integrated into the bottom surfaces of each well to measure cell index based on impedance using the xCELLigence System (Roche, Indianapolis, IN, USA). First, 100 µL of cell culture media was first added to each well at 37 °C for 30 min to equilibrate the conductors to the media. Then, 8 k, 4 k, 2 k and 1 k of cells were added per well, and the program was set to conduct readings every 30 min for 95 h. Cell migration was measured by using Culture-Inserts (IBIDI, Martinsried, Germany). The culture-inserts were transferred to six-well plates, and the cells were seeded at a density of 7 × 10^4^/mL in each well of culture-inserts and cultured in phenol red-free complete medium. After 24 h of incubation, the culture-inserts were removed, and a 400 μm cell-free gap was created. Cell migration was observed with a Leica DMI6000 inverted microscope equipped with a Hamamatsu ORCA-ER camera using LEICA N PLAN 10×, /20× /0.25 objectives, and recorded every 20 min for 23 h. Image processing was carried out using the Leica (LAS) and Adobe Photoshop software. Quantification of the cell surface was performed using the multiwavelength cell-scoring module of Metamorph Offline software and the WimScratch Wound Healing Module (WIMASIS, Munich, Germany) program. Transwell invasion assays were performed using Transwell chambers (24 well, 8 μm pore size; Cell Biolabs, San Diego, CA, USA) coated with basement membrane. The cells were allowed to grow to subconfluency and were serum-starved for 24 h. After detachment with trypsin, the cells were washed with PBS, resuspended in serum-free medium and 5 × 10^4^ cells were added to the upper chamber. The complete medium was added to the bottom well of the chamber. The cells that had not migrated were removed from the upper face of the filters using cotton swabs, and the cells that had migrated to the lower face of the filters were fixed with 3.7% formaldehyde, permeabilized with 100% methanol and stained with 0.05% crystal violet. Invasive cells were solubilized with 1% SDS at room temperature for 30 min, and quantified at OD 590 nm.

### 4.6. Immunofluorescence Microscopy

Cells grown on coverslips were fixed in 4% paraformaldehyde/phosphate-buffered saline at room temperature for 15 min and then permeabilized by incubation in 0.01% Triton X-100 in PBS. After fixation, the coverslips were blocked with 3% BSA in PBS for 45 min and incubated with primary antibody for 1 h followed by appropriate Alexa488- and Alexa594-conjugated secondary antibodies or rhodamine phalloidin (Molecular Probes, Eugene, Oregon USA) for 1 h at room temperature. The coverslips were counterstained with DAPI (Sigma-Aldrich) and mounted with the anti-fade agent Prolong Gold (Molecular Probes). Images were acquired with a Leica DM6000B microscope. Data were processed with Adobe Photoshop CS4 (Adobe Systems, San José, CA, USA).

### 4.7. RNA Isolation and RT-qPCR

Total RNA was isolated using TRIzol (Invitrogen, Life Technologies, Karlsrube, Germany), treated with TURBO DNase (Invitrogen), and reverse-transcribed with a mix of oligo(dT)_20_ and random hexamers using SuperScript III reverse transcriptase (Invitrogen). RT-qPCR was performed using iTaq Universal SYBR Green Supermix (Bio-RAD) and analyzed on an Applied Biosystems 7500 Fast-Real-Time PCR System. Expression levels of mRNAs were normalized to *Hprt* levels. According to the published data on miRNAs expression in NMuMG cells by TGF-β treatment [39], we used the miR-425-5p gene to normalize miRNAs expression.

### 4.8. Poly(A)^+^ and Poly(A)^−^ RNA Separation

Separation of Poly(A)**^+^** and Poly(A)**^−^** RNA was essentially performed as described [40]. DNAse-treated total RNA was incubated with oligo(dT) magnetic beads (Dynabeads, Invitrogen) to isolate poly(A)**^+^** RNAs, which were bound to the beads, or poly(A)-RNAs, which were present in the flow through after incubation. Selection with oligo (dT) magnetic beads was performed three times to ensure pure poly(A)**^+^** and poly(A)**^−^** populations. The poly(A)**^−^** RNA population was further processed with the Ribo-Zero magnetic kit (Epicentre, Epicentre Biotechnologies, Madison, WI, USA) to deplete most of the abundant ribosomal RNA.

### 4.9. In Vitro Poly(A) Tailing and RACE

To determine the 3′ end(s) of the *lnc-NR6a1* precursor, in vitro poly(A)**^−^** tailing of total RNA was carried out with the poly(A) Polymerase Tailing Kit (Epicentre) according to the manufacturer’s protocol. 3′ and 5′ RACE were performed using the FirstChoice RLM-RACE kit (Invitrogen) following the manufacturer’s protocol. The RT-PCR primer sequences are listed in Appendix A.

### 4.10. RNA Isolation from Mouse Tissues

Organs from normal C57BL/6 mice were dissected and immediately frozen in liquid nitrogen and stored at −80 °C. To isolate RNA, 40–60 mg of frozen tissue specimens were cut in small pieces and transferred to a tube containing 1 mL of TRIzol Reagent (Invitrogen). Next, they were mechanically homogenized using a Pro250 homogenizer (PROScientific, Oxford, CT, USA) and total RNA was extracted according to the manufacturer’s instruction.

### 4.11. Cell Fractionation

The separation of cytosolic, chromatin and soluble nuclear fractions was carried out as previously described [41]. RNA was extracted from each fraction using a TRIzol reagent (Invitrogen).

### 4.12. RNA-Seq Analysis

Total RNA was isolated using TRIzol (Invitrogen), depleted of DNA by DNase treatment (TURBO DNase, Invitrogen). Libraries were prepared with the TruSeq Stranded mRNA kit (Illumina, San Diego, CA, USA) and sequencing was performed with a Novaseq system (Illumina) with 75 bp single-end reads with the Genomic Unit of CABIMER (Sevilla, Spain). Two biological replicates for each condition were sequenced. Reads were aligned to mouse genomes (GRCm38/mm10 and NCBI37/mm9) using the *subjunc* function from the Rsubread package. The downstream analysis was performed on .bam files with duplicates removed using the samtools (v0.1.19) rmdup command. Differential gene expression analysis and statistics were performed using the DESeq2 Bioconductor. Differentially expressed genes with *p*-adjusted values < 0.05 and Log_2_FC > 0.5 (upregulated genes) or Log_2_FC < −0.5 (downregulated genes) were selected for further analysis. Gene ontology (GO) functional categories were analyzed using DAVID Bioinformatics Resources. Non-adjusted *p* values were used to determine the enrichment significance.

### 4.13. ChIP Sequencing

ChIP assays and RNA immunoprecipitation were performed as previously reported [42]. NMuMG cells were crosslinked in 1% formaldehyde for 10 min at room temperature followed by the addition of glycine (125 mM final concentration) for 5 min. Nuclei were isolated using lysis buffer 1 (5 mM Pipes pH 8, 85 mM KCl, 0.5% NP-40, and complete protease inhibitor cocktail (Roche, Hoffmann-La Roche Inc., Nutley, NJ, USA)) and were lysed using lysis buffer 2 (1% SDS, 10 mM EDTA, 50 mM Tris–HCl pH 8.1, and complete protease inhibitor cocktail (Roche)). Chromatin was sheared into an average size of 500 bp by eight pulses of 30 s (30 s pause between pulses) at 4 °C in the water bath sonicator Bioruptor (Diagenode, Liège, Belgium).Thirty micrograms of chromatin were diluted 1:10 in IP buffer (0.01% SDS, 1.1% Triton X-100, 1.2 mM EDTA, 16.7 mM Tris–HCl pH 8.1, 167 mM NaCl, 1% sodium deoxycholate) and incubated overnight at 4 °C in rotation with 5 µg of monoclonal antibody anti-Lamin A/C (sc-376248; Santa Cruz) or anti-rabbit IgG control antibody (Sigma). Complexes were collected using Protein-A Dynabeads pre-blocked with BSA (New England Biolabs) and transfer RNA (Roche) and then washed with wash buffer 1 (0.1% SDS, 1% Triton X-100, 2 mM EDTA, 20 mM Tris–HCl pH 8.1, and 150 mM NaCl, 1% sodium deoxycholate), wash buffer 2 (0.1% SDS, 1% Triton X-100, 2 mM EDTA, 20 mM Tris–HCl pH 8.1, 500 mM NaCl, 1% sodium deoxycholate), wash buffer 3 (0.25 M LiCl, 1% NP-40, 1% sodium deoxycholate, 1 mM EDTA, 10 mM Tris–HCl pH 8.1), and twice with TE 1× buffer (10 mM Tris–HCl pH 8.0, 1 mM EDTA). The complexes were eluted twice from the beads with elution buffer (1% SDS in TE 1× buffer) by incubating 10 min at 65 °C. The eluates and the inputs were incubated overnight at 65 °C for de-crosslinking and treated with Proteinase K for 1 h at 37 °C. Chipped DNA was purified with phenol: chloroform extraction followed by ethanol precipitation and resuspended in Milli-Q water. The sequencing was performed with the Illumina HiSeq2000 system (Illumina) with 75 bp paired-end reads at the Genomic Unit of CABIMER (Sevilla, Spain). ChIP-seq reads for each condition were independently aligned against the MGSCv37/mm9 mouse reference genome using the align function from the Bioconductor Rsubread package (v1.28.1) with type = 1, TH1 = 2 and unique = TRUE parameters. Then a differential binding analysis was performed to identify Lamin Associated Domains (LADs) using the csaw (v1.12.0) and edgeR (v3.20.9) packages. Briefly, aligned reads were counted in bins of 25 kb along the genome using the windowCounts function and read coverage of each bin was normalized to library size using calcNormFactors. To perform differential binding analysis for each window using input as the control, a sequential number of functions were used: estimateDisp, glmFit and glmLRT. Finally, enriched windows were merged as they were overlapping or adjacent using the mergeWindows function.

### 4.14. RNA Immunoprecipitation

RNA immunoprecipitation was performed after the formaldehyde crosslinking of cells. Cells were resuspended in the RIPA buffer and incubated for 30 min at 4 °C and next homogenized by sonication (Digenode’s Bioruptor, 10 min, high power setting, 15 s on/30 s off). Precleared extracts were incubated overnight with either antibody anti-Lamin A/C, anti-γ-Catenin, anti-ENO1, anti-ALDOA, anti-GAPDH or IgG (Sigma). Complexes were captured using Protein-A Dynabeads, washed four times for 5 min at 4 °C with RIPA buffer and resuspended in Tris-HCl pH 7.0, 5 mM EDTA, 10 mM DTT and 1% SDS and next incubated at 70 °C for 45 min to reverse the crosslinks. RNA was extracted using TRIzol (Invitrogen) according to the manufacturer’s protocol, reversed transcribed and subjected to PCR.

### 4.15. CHIRP

Chromatin isolation by RNA purification (ChIRP) was essentially performed as described [43]. Anti-sense oligo probes were designed at http://singlemoleculefish.com (accesssed on 4 August 2022) (Appendix A). Two negative controls were used in this assay: *LacZ* antisense DNA probes were used in control cells, while *lnc-Nr6a1-1* probes were used in *Lnc-Nr6a1*-depleted cells. Input and purified DNA were sequenced using the Illumina NextSeq 500 system (Illumina) with 75 bp paired-end reads at the Genomic Unit of CABIMER (Sevilla, Spain). ChIRP-seq reads for each condition were independently aligned against the MGSCv37/mm9 mouse reference genome using the align function from the Bioconductor Rsubread package (v1.28.1) with type = 1, TH1 = 2 and unique = TRUE parameters. Peak calling was performed using macs2 callpeak (v2.1.2) using input as the control data and 2620345972 bp as the effective genome size. Then, peaks were filtered using −log10(FDR) ≥ 10 criteria and those peaks that overlap with the blacklisted region of the genome were removed using bedtools subtract.

### 4.16. CRISPR-SAM-Mediated Activation of Lnc-Nr6a1

CRISPR activation of the *Lnc-Nr6a1* expression was achieved by the Synergistic Activation Mediator (SAM) using the three-vector system from Addgene (Addgene, Watertown, MA, USA): lenti dCAS-VP64 (Addgene plasmid #61425), lenti MS2-P65-HSF1 v2 (Addgene plasmid #89308) and lenti MS2-sgRNA (Addgene plasmid #61427) (Addgene. NMuMG cells were infected with dCAS-VP64 and MS2-P65-HSF1 lentiviruses and selected for 7 days with blasticidin (5 µg /mL) and hygromycin (200 µg/mL). Gene-specific sgRNA targets were designed using online tools provided by IDT (Integrated DNA Technologies) and CRISPOR (http://crispor.org, accessed on 4 August 2022). Guides (Appendix A) were designed within –300/+100 bp of the transcription start site (TSS) of the *Lnc-Nr6a1* gene, with most guides within the proximal promoter (∼190 bp of the TSS), and they were cloned into the lenti sgRNA(MS2) plasmid. Cells were infected with lenti sgRNA(MS2) lentiviruses expressing *Lnc-Nr6a1*-targeting sgRNA and selected with 400 µg/mL zeocin. The expression of *lnc-Nr6a1* isoforms and miR-181a2 and miR-181b2 was then quantified with qRT-PCR.

### 4.17. CRISPR/Cas9-Mediated Deletion of Lnc-Nr6a1

The Alt-R CRISPR-Cas9 System (IDT) [44] was used to generate *Lnc-Nr6a1*-depleted NMuMG cells, according to the manufacturer’s protocol. Upstream and downstream guides were designed within −600/−500 bp and +1000/+1100, respectively, of the TSS. Guide sequences are shown in Appendix A. Successfully transfected cells were sorted by fluorescent-activated cell sorting (FACS) and individually distributed into 96-well plates. After 10–14 days, cells in each well were resuspended in 200 µL of media and 100 µL were plated into two separate 96-well flat-bottom plates. One of them was kept to allow clones to grow and the other one to screen each clone for deletion. Genomic DNA was isolated using the Purelink Genomic DNA Kit (Invitrogen) according to the manufacturer’s protocol. Deletions were analyzed by PCR and deletions amplicons were sequenced to identify the precise deletion. RNA was isolated from monoallelic and biallelic deletion clones and analyzed by RT-qPCR to quantify the *Lnc-Nr6a1* expression.

### 4.18. iDRIP

Experiments were performed as previously described [45]. Briefly, cells were crosslinked with 0.4 Jcm^−2^ of UV_254nm_ in cold PBS, harvested by scraping and pelleted. Crosslinked cells were resuspended in DNase I digestion buffer containing 50 mM Tris pH 7.5, 0.5% NP-40, 0.1% sodium lauroyl sarcosine, protease inhibitors, RNasin, 10 mM vanadil ribonucleoside and 600 U Turbo DNAse and incubated at 37 °C for 15 min with rotation. Next, lysates were further supplied to the final concentrations of 1% sodium lauroyl sarcosine, 0.1% sodium deoxycholate, 0.5 M lithium chloride, 25 mM EDTA and 25 mM EGTA and incubated at 37 °C for 5 min. The lysates were spun at 2500 rpm 10 min and the supernatans were collected. The pellets were resuspended in lysis buffer containing 50 mM Tris pH 7.5, 0.5 M lithium chloride, 1% NP-40, 1% sodium lauroyl sarcosine, 0.1% sodium deoxycholate 20 mM EDTA and 20 mM EGTA, incubated on ice for 10 min, heated to 65 °C for 5 min, and spun at room temperature; and the supernatant combined with the previous supernatants. The combined supernatants were pre-cleared by incubation with the washed MyOne streptavidin C1 magnetic beads (ThermoFisher) at room temperature for 30 min with rotation. The Streptavidin beads were then magnetically separated from the lysate samples using a Dynamag magnet (Life Technologies). Capture 3′biotin-TEG oligonucleotides and beads were mixed and incubated in 2× binding buffer containing 10 mM Tris-HCl pH 7.5, 1 mM EDTA and 2 M NaCl at room temperature for 20 min, washed with 1× binding buffer and then resuspended in lysis buffer. Pre-cleared lysates and oligonucleotides-conjugated beads were preheated to 65 °C for 5 min, mixed and incubated at 65 °C for another 15 min in the hybridization oven, followed by slowly reducing the temperature at 37 °C and 1 h incubation. Beads with captured hybrids were washed three times at 37 °C in wash buffer1 containing 50 mM Tris, pH 7.5, 0.3 M LiCl, 1% SDS, 0.5% NP-40, 1 mM DTT and protease inhibitors, followed by treatment at 37 °C for 10 min with 20 U of Turbo DNase I in DNase digestion buffer supplemented in 0.3 M LiCl, protease inhibitors and RNasin. After washing two more times at 37 °C with wash buffer1, the lysates were further washed at 37 °C for 5 min in wash buffer2 containing 1% SDS, 1 mM DTT, 5 mM EDTA, 150 mM NaCl and 1 mM PMSF. The proteins were eluted at 70 °C for 5 min in elution buffer containing 10 mM Tris, pH 7.5 and 1 mM EDTA. The eluted proteins eluted were lyophilized and stored at −20 °C until processing for mass spectrometry. As the negative control, *LacZ* (non-mammalian) capture probes were used. The Sequences of oligonucleotides are listed on Appendix A.

### 4.19. Protein Identification Using Mass Spectrometry

Proteins were in-gel digested as described [46]. Briefly, the eluted and dried-down proteins from the iDRIP experiment were subjected to SDS-PAGE gel and the whole proteome band upon entering the resolving gel was subjected to trypsin disgestion. The resulting peptides were analyzed by nanoliquid chromatography coupled to mass spectrometry, using a Q-Exactive HF mass spectrometer (ThermoFisher), for protein identification and quantification by spectral counting [47]. Peptide identification from MS/MS data was performed using the probability ratio method [48]. False discovery rates (FDR) of peptide identifications were calculated using the refined method [49,50]; 1% FDR was used as criterion for peptide identification.

### 4.20. Adhesion Assays

Matrix adhesion assays were performed as described previously [51]. Briefly, 12-well plates were blocked with 1% bovine serum albumin for 1 h at 37 °C. NMuMG cells (3 × 10^5^) were added to the wells and then incubated for 1, 4, 6 and 8 h at 37 °C. After three washings with PBS, the attached cells were fixed and stained with crystal violet. After extensive washing with PBS, bound dye was solubilized with 1% SDS, and the absorbance was read at a wavelength of 590 nm. Cell–cell adhesion assays were performed by resuspending NMuMG cells in serum-free medium and incubated with 5 μm calcein-AM dye for 30 min at 37 °C. After incubation, non-incorporated calcein-AM was removed by three washes with serum-free medium. 4 c10^4^ calcein-AM-labeled cells were added to a confluent monolayer of unstained NMuMG cells grown in a 96-well plate. After 2 h incubation at 37 °C, unbound calcein-AM-labeled cells were removed by washing with PBS. Adherent cells were dissociated from the culture plates and resuspended in PBS, and flow cytometry-based quantification of cells were carried out by FACS analysis. All adhesion experiments were done in triplicate and repeated at least three times.

### 4.21. Anoikis Induction and Flow Cytometry

Anoikis was assayed by plating cells on polyHEMA-coated plates. A solution of 120 mg mL^−1^ polyHEMA (Sigma) in 100% ethanol was made and diluted 1:10 in 95% ethanol; 0.95 µL mm^−2^ of this solution was overlaid onto 35-mm wells and left to dry in a heated dryer system for 12 h. Before use, wells were washed twice with PBS and once with DMEM. In all, 3.5 × 10^5^ cells of each line, suspended in 2 mL DMEM containing methylcellulose 1% were incubated in the polyHEMA-coated wells for 18–24 h in a humidified (37 °C, 5% CO_2_) incubator. Cells were harvested and resuspended in 0.3 mL of PBS. Cells were fixed with ice-cold ethanol (70% final concentration), resuspended in 300 µL solution of 50 μg/mL propidium iodide and 250 μg/mL DNase-free RNase in PBS, and incubated for 30 min at 37 °C. For flow cytometric analysis, at least 10,000 cells were evaluated using a FACSCalibur™ cytometer. Cell-cycle distribution and Sub-G_1_ fraction were determined and quantified using the CellQuest-Pro software.

### 4.22. Seahorse Assay

XFe24 Extracellular Flux Analyzer (Agilent, Santa Clara, CA, USA) was used to determine glycolysis metabolism. Cells were plated in 24-well (1.5 × 10^4^ cells/well) Seahorse Assay plates, grown in Dulbecco’s modified Eagle’s medium containing 10% fetal bovine serum, L-glutamine, antibiotics and insulin (10 µg/mL) and incubated overnight at 37 °C in a 5% CO_2_ humidified, 95% air incubator. The sensor cartridge was hydrated adding 500 µL of XF Calibrant Solution (Agilent) at 37 °C in a CO_2_-free incubator overnight. GlycoPER was determined using the Agilent Seahorse XF Glycolytic Rate Assay Kit (Agilent, Cat No: 1033344). Data were analyzed by software Seahorse XFe (Agilent). All measurements were normalized with total protein concentration on each well. Each assay was run with 6 replicates per each condition.

### 4.23. Analysis of Tumor Data Sets

Data on co-expression analysis for *miR181A2HG* (human *Lnc-Nr6a1*) and *hsa-miR181a2-3p* levels from 9 tumor types (4179 samples) were obtained from the TCGA project via the Genomic Data Commons Data Portal. Nomenclature, cancer full name and sample number are shown in Appendix A. GEPIA data were used to obtain *miR181A2HG* (human *Lnc-Nr6a1*) expression profiles of 18 types of human cancer and the corresponding adjacent normal tissues.

### 4.24. Statistical Analysis

Data were analyzed by one-way ANOVA and Student’s *t*-test comparison, using GraphPad Prism 5 (Graphpad Software, La Jolla, CA, USA). Significant *p*-values are indicated with asterisks as follows: * *p* < 0.05, ** *p* < 0.01, and *** *p* < 0.001. The probabilities of overlapping genes were calculated using the hypergeometric distribution using the Keisan Online Calculator (http//keisan.casio.com/exec/system/1180573201, accessed on 4 August 2022).

## Figures and Tables

**Figure 1 ncrna-08-00062-f001:**
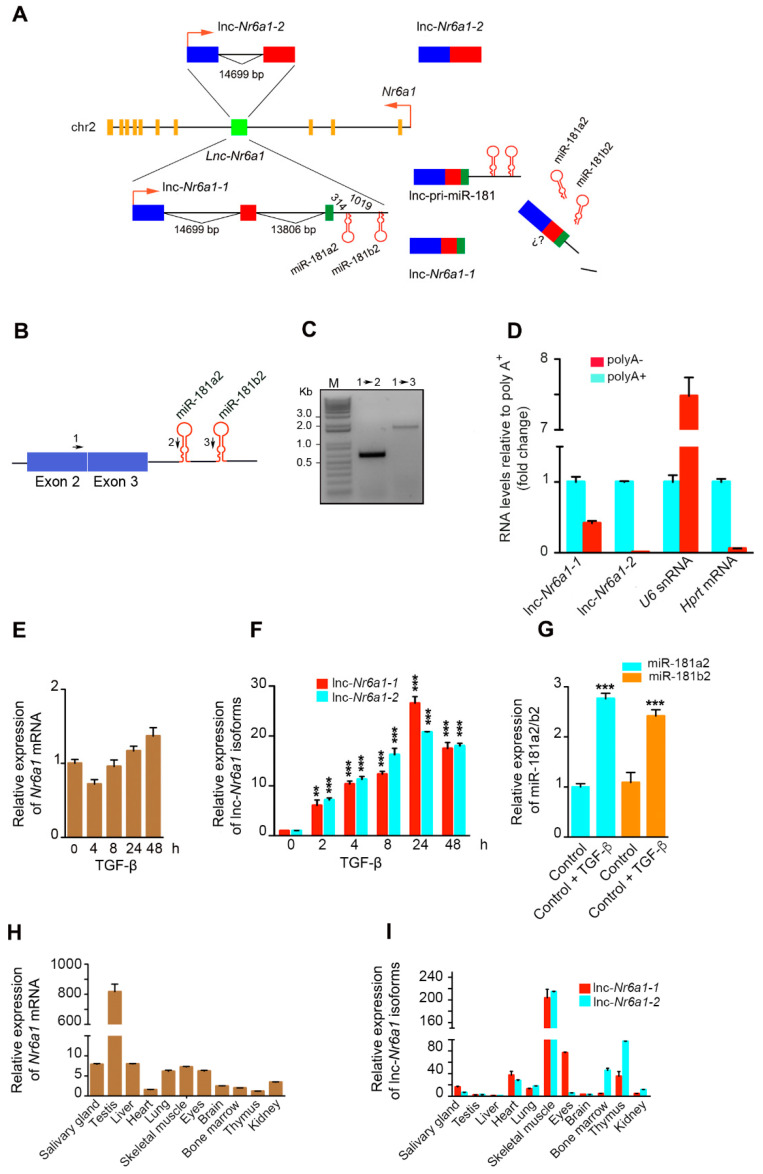
Genomic context and expression profiles of *Lnc-Nr6a1* and *Nr6a1* genes. (**A**) Schematic illustration of the genomic location and transcript structure of the *Lnc-Nr6a1* gene embedded within the third intron of the *Nr6a1* gene. The annotated nascent and processed transcripts of *Lnc-Nr6a1* (*lnc-Nr6a1-1* and *lnc-Nr6a1-2*) and lnc-pri-miR-181a2/b2 are shown. The red arrows indicate the transcription orientation. (**B**) Schematic representation of spliced lnc-pri-miR-181a2/b2 transcript and (**C**) electrophoretic gel showing PCR products generated with a forward primer located in exon2 and reverse primers located in miR-181a2 and mir181b2, respectively. (**D**) Rate of poly(A**^−^**) RNA relative to poly(A**^+^**) RNA for the indicated transcripts, quantified by RT-qPCR. (**E**) Relative levels of *Nr6a1* mRNA in cells treated with TGF-β at the indicated times. Experiments were performed in triplicate samples and error bars represent S.D. (**F**) Relative levels of *lnc-Nr6a1* isoforms in cells treated with TGF-β at the indicated times. (**G**) Expression of miR-181a2/b2 in cells treated with TGF-β for 24 h. Experiments were repeated twice in triplicate samples; error bars represent S.D. (**H**,**I**) relative expression of *Nr6a1* mRNA and *lnc-Nr6a1* isoforms, respectively, in a broad range of adult mouse tissues. Results are presented relative to a fixed reference sample (thymus and liver in (**H**,**I**), respectively). Experiments were performed in triplicate samples; error bars represent S.D. *** *p* < 0.001, ** *p* < 0.01 by two-tailed Student’s *t*-test.

**Figure 2 ncrna-08-00062-f002:**
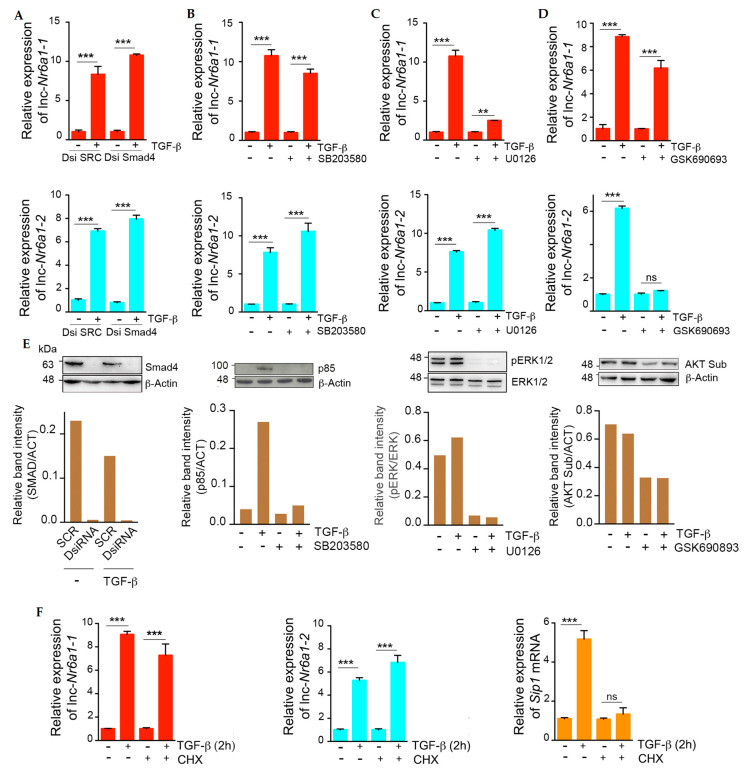
Signaling pathways involved in *lnc-Nr6a1* isoforms upregulation by TGF-β. (**A**) NMuMG cells were transfected with DsiRNAs against Smad4 (DsiSmad4) or scramble DsiRNA (SRC) and treated with TGF-β for 2 h. Levels of *lnc-Nr6a1-1* (top) and *lnc-Nr6a1-2* (bottom) were measured by RT-qPCR analysis. Experiments were performed in triplicate samples; error bars represent S.D. (**B**) Effect of p38 kinase inhibitor, SB203580, on *lnc-Nr6a1* isoforms upregulation. Cells were pre-treated with 10 µM inhibitor for 1 h and next treated with TGF-β for 2 h where indicated. Levels of *lnc-Nr6a1-1* (top) and *lnc-Nr6a1-2* (bottom) were measured by RT-qPCR analysis. Experiments were performed in triplicate samples; error bars represent S.D. (**C**,**D**). Effect of MAP kinase inhibitor U0126 and AKT inhibitor GSK690693, respectively, on *lnc-Nr6a1* isoforms upregulation. NMuMG cells were pre-treated with 10 µM inhibitor for 1 h prior to addition of TGF-β for 2 h where indicated. Levels of *lnc-Nr6a1-1* (top) and *lnc-Nr6a1-2* (bottom) were measured by RT-qPCR analysis. Experiments were performed in triplicate samples; error bars represent S.D. (**E**) Representative immunoblot (top) and densitometry analysis (bottom) for SMAD4, p38 kinase (phosphorylated p85 ribosomal kinase, p85), MAP kinase (phosphorylated ERK1/2, pERK1/2) and AKT kinase (AKT phosphorylated substrate, AKT Sub) are shown. β-actin was used for protein-loading normalization in SMAD4, p38 kinase and AKT kinase immunoblots. Total ERK1/2 were used as the protein-loading control in the ERK1/2 immunoblot. (**F**) Requirement of de novo protein synthesis for *lnc-Nr6a1* isoforms upregulation by TGF-β. NMuMG cells pre-treated with 5 µM of cycloheximide (CHX) for 1 h were stimulated with TGF-β for 2 or 24 h. Expression levels of *lnc-Nr6a1* isoforms and SIP1 were measured by RT-qPCR analysis. Experiments were performed in triplicate samples; error bars represent S.D. *** *p* < 0.001, ** *p* < 0.01, ns, not significant by two-tailed Student’s *t*-test.

**Figure 3 ncrna-08-00062-f003:**
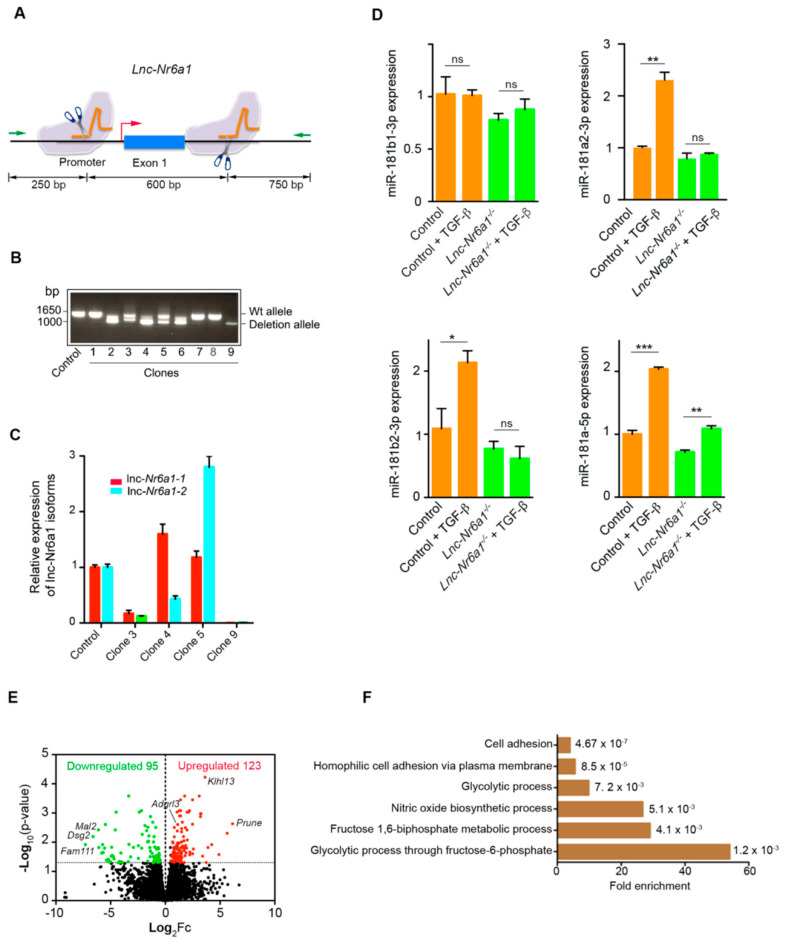
Deletion of a *Lnc-Nr6a1* fragment in NMuMG cells and expression profile. (**A**) Scheme showing the 600 bp deleted fragment of *Lnc-Nr6a1* deletion in NMuMG cells using CRISPR-Cas9 technology. Sites of cleavage by Cas9 in the promoter and first exon regions are indicated; primers (green arrows) for clones analysis are shown (**B**) Representative gel identifying non-deletion (control, clones 1, 7 and 8) monoallelic (clone 2, 3, 4, 5 and 6) and biallelic clones (clone 9). (**C**) Expression levels of *lnc-Nr6a1* isoforms in different clones were measured by RT-qPCR analysis. (**D**) Expression of miR-181a/b species in control and *Lnc-Nr6a1*-depleted cells after 24 h of TGF-β treatment or untreated. Experiments were repeated twice in triplicate samples; error bars represent S.D. *** *p* < 0.001, ** *p* < 0.01, * *p* < 0.05, ns, not significant by two-tailed Student’s *t*-test. Y-axis of last panel (down right) is labeled 181a-5p because the isoforms miR-181a1-5p and miR-181a2-5p are identical. (**E**) Volcano plot for differentially expressed genes in *Lnc-Nr6a1*-depleted cells versus control cells. Genes upregulated with more than a 1.5-fold change with a *p*-value < 0.05 are depicted in red boxes and those downregulated with identical fold change and *p*-value are in green boxes. (**F**) Gene ontology enrichment analysis of downregulated genes in *Lnc-Nr6a1*-depleted cells.

**Figure 4 ncrna-08-00062-f004:**
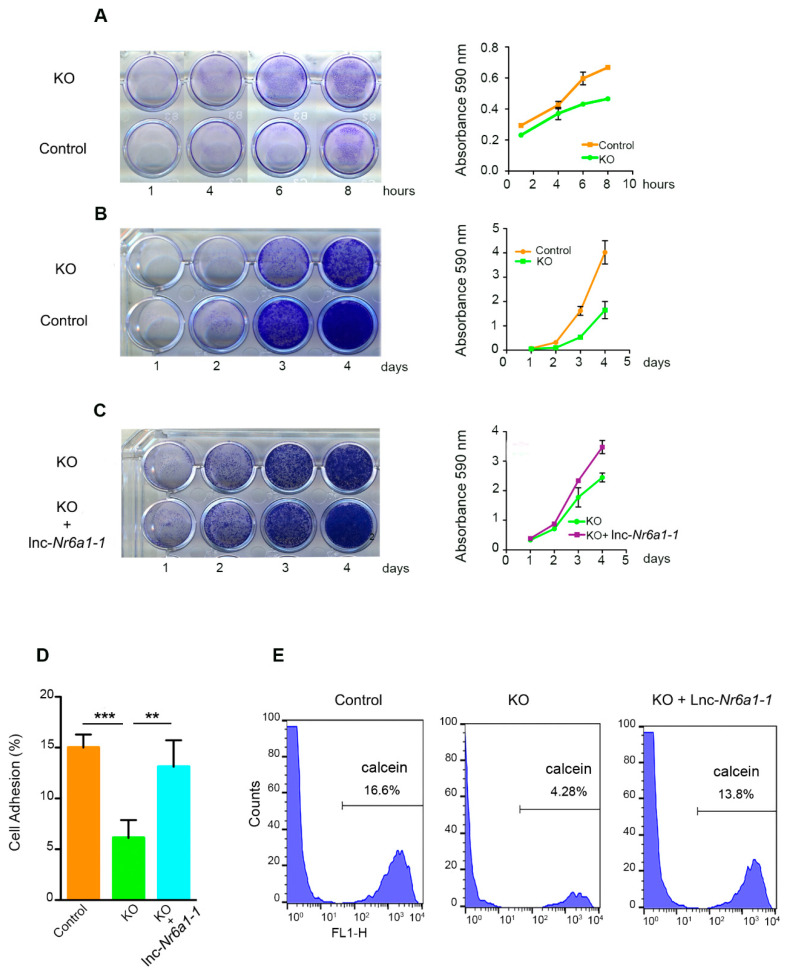
*Lnc-Nr6a1* depletion affects cell adhesion. Scanned 60-mm Petri dishes with fixed colonies subjected to crystal violet staining assays, (**A**,**B**) control cells and *Lnc-Nr6a1*-depleted cells, (**C**) *Lnc-Nr6a1*-depleted cells and *Lnc-Nr6a1*-depleted cells overexpressing *lnc-Nr6a1-1* isoform. Cells were plated on dishes and incubated for indicated times. Stained adherent cells were solubilized with 1% SDS at room temperature for 30 min and quantified at OD 590 nm. Experiments were performed in triplicate, and data are expressed as the mean ± S.D. from three independent experiments. (**D**,**E**) Flow cytometric analysis of control, *Lnc-Nr6a1*-depleted and *Lnc-Nr6a1*-depleted cells, overexpressing *lnc-Nr6a1-1* calcein-labeled cells. Labeled cells were added to a confluent monolayer of unstained NMuMG cells incubated for 2 h and adherent cells quantified by flow cytometry analysis. The bar graph shows mean ± S.D. of six independent experiments, *** *p* < 0.001, ** *p* < 0.01 via two-tailed Student’s *t*-test.

**Figure 5 ncrna-08-00062-f005:**
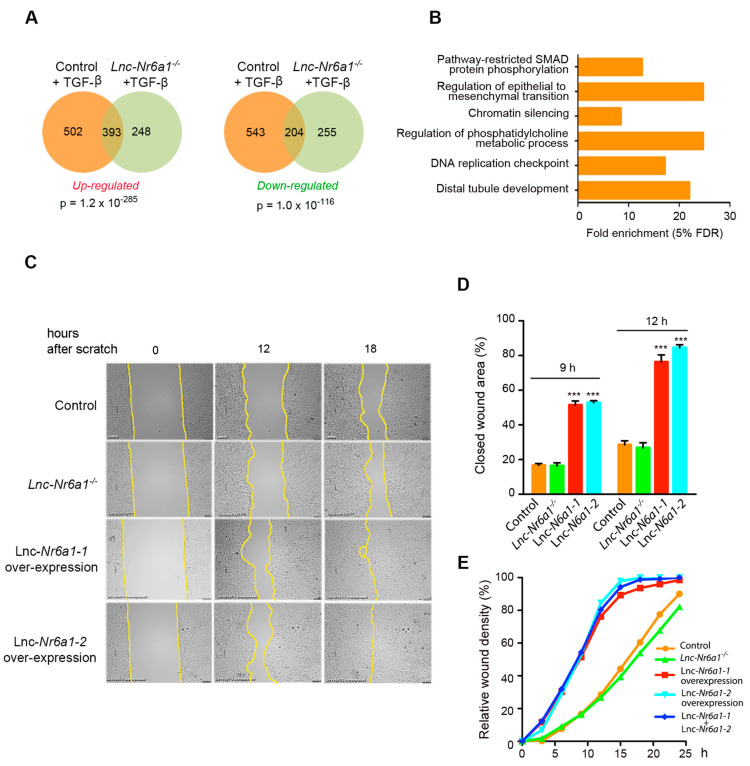
*Lnc-Nr6a1* depletion affects gene expression changes by TGF-β. (**A**) Venn diagrams showing the intersection of upregulated (left panel) and downregulated genes (right panel) in control and *Lnc-Nr6a1*-depleted cells treated with TGF-β for 2 h. Probability of overlapping based on hypergeometric distribution is provided. (**B**) Significant Gene Ontology annotation enrichments of genes affected by TGF-β treatment independently of the presence or absence of *Lnc-Nr6a1*. (**C**) Migration of control, *Lnc-Nr6a1*-depleted, *lnc-Nr6a1-1*-upregulated and *lnc-Nr6a1-2*-upregulated NMuMG cells were tested in wound-healing assays. Cells were imaged at 10 min intervals for 26 h. Frames of the movie at 12 and 18 h are shown. (**D**) Percentage of surface covered using frames at 9 and 12 h. Experiments were performed in triplicate samples. Values represent the percentage of wound closures. *p*-values are indicated for comparison. Error bars represent S.D. *** *p* < 0.001. (**E**) A representative wound-density quantification of the time-lapse video frames from different cells using Wimasis image analysis.

**Figure 6 ncrna-08-00062-f006:**
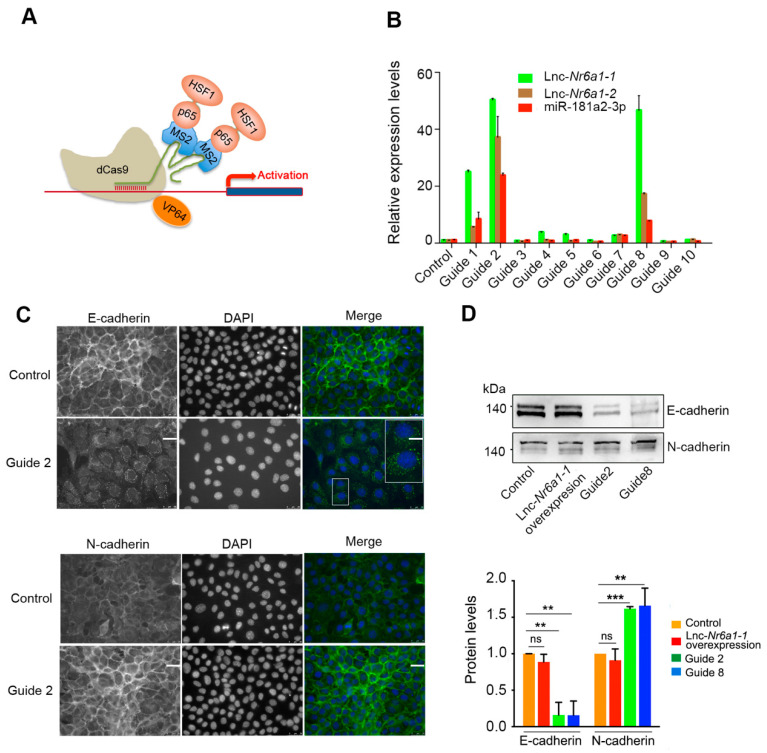
Over-expression of *Lnc-Nr6a1*. (**A**) Diagram of CRISPR SAM-CRISPR/Cas9 activation system. Nuclease-dead Cas9 (dCas9), fused to the transcriptional activator VP64, complexes with the CRISPR guide RNA. MS2 bacteriophage coat protein is fused to the p65 subunit of NF-kappaB and the activation domain of human heat-shock factor 1 (HSF1). MS2 coactivator proteins bind to MS2 RNA aptamer in the gRNA and with VP64 synergistically activate gene expression. (**B**) Expression levels of *lnc-Nr6a1* isoforms and miR-181a2-3p in cells lines with different sgRNAs (within the region from −285 to +105bp, distance to TSS). *Hprt* and miR-425-5p genes were used to normalize expression of *lnc-Nr6a1* isoforms and mR-181a2/b2, respectively. (**C**) Representative image from immunofluorescence staining of changes in the expression of marker proteins. Cells were stained with antibodies against the epithelial marker E-cadherin, mesenchymal marker N-cadherin and 4,6-diamidino-2-phenylindole to visualize nuclei. Scale bar, 25 µM. (**D**) Representative immunoblotting of E-cadherin and N-cadherin in control, *lnc-Nr6a1-1*-upregulated and *Lnc-Nr6a1*-upregulated cells (guide 2 and guide 8). Quantification of E-cadherin and N-cadherin in three independent experiments is shown in panel 6D (down). Error bars represent S.D. *** *p* < 0.001, ** *p* < 0.01, ns, not significant by two-tailed Student’s *t*-test. Protein-loading normalization was performed by measuring total protein directly on the membrane using the criterion stain-free gel imaging system. (**E**) The migratory capacity of the control *lnc-Nr6a1-1*- and *Lnc-Nr6a1*-upregulated cells were tested in wounding-healing assays. Cells were imaged at 10 min intervals for 24 h. The frames of the movie at 12 and 18 h are shown. (**F**) The frames at 12 and 18 h were used to estimate the percentage of surface covered by the cells. Experiments were performed in triplicate samples. Values represent the percentage of wound closure. Error bars represent S.D. *p*-values for comparisons are indicated. *** *p* < 0.001 by two-tailed Student’s *t*-test. (**G**) Representative images of invasion of the control *lnc-Nr6a1-1*- and *Lnc-Nr6a1*-upregulated (guide 2) cells analyzed in Transwell invasion assays. After 24 h, cells were fixed and stained with crystal violet. Invasive cells were solubilized with 1% SDS at room temperature for 30 min and quantified at OD 590 nm. Histograms (panel 6G, down right) represent the mean ± S.D. from three independent experiments. ** *p* < 0.01, * *p* < 0.05 by two-tailed Student’s *t*-test. (**H**) Control, *lnc-Nr6a1-1*- and *Lnc*-*Nr6a1*-upregulated (guide 2 and guide 8) cells were detached and suspended on poly-HEMA coated plates for 18 h in presence of 0.5% methylcellulose. Percentage of sub-G1 cells was determined by flow cytometry. Results are shown as the averages ± standard errors of the means from six independent experiments and were analyzed by one-way ANOVA, followed by the Bonferroni post-test for significance versus control cells. *** *p* < 0.001, ** *p* < 0.01, ns not significant.

**Figure 7 ncrna-08-00062-f007:**
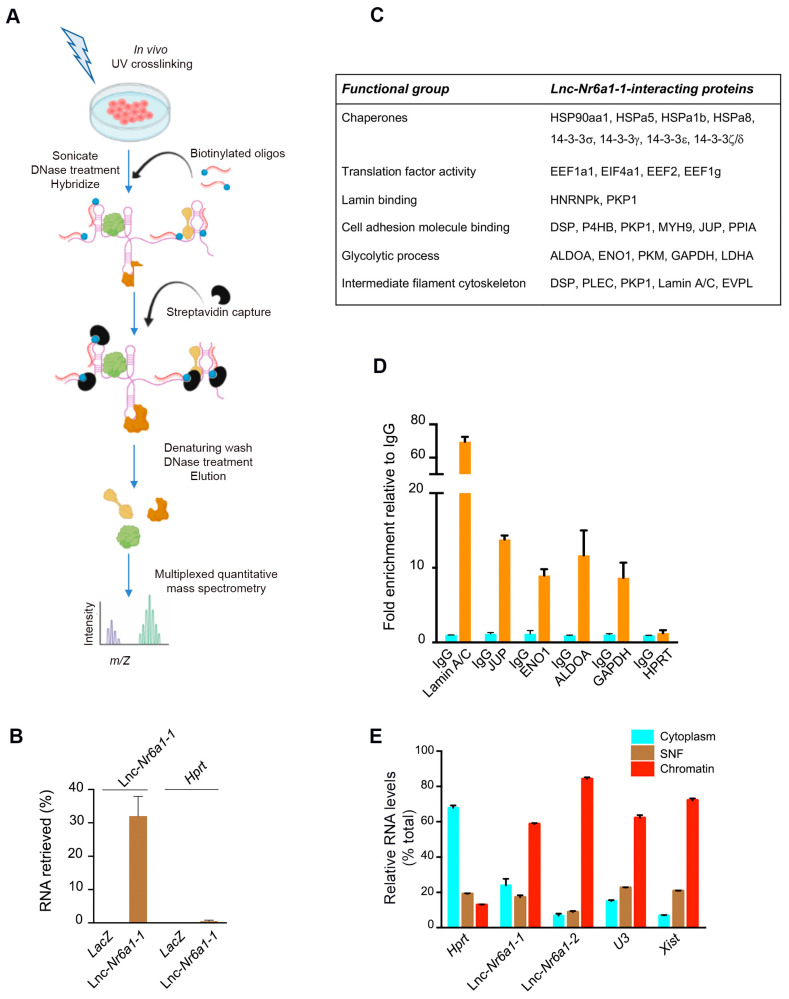
iDRIP-MS reveals direct *lnc-Nr6a1-1*-interacting proteins. (**A**) A schematic overview of the iDRIP-MS method. (**B**) Retrieval of RNA by iDRIP using *lnc-Nr6a1-1* and *LacZ* probes measured by RT-qPCR. Efficiency of *lnc-Nr6a1-1* pulldown was determined by comparing to a standard curve generated using serial dilutions of input. Error bars S.D. (**C**) Selected *lnc-Nr6a1-1*-interacting proteins identified by iDRIP are grouped into functional classes. (**D**) Physical association of *lnc-Nr6a1-1* with Lamin A/C, g-catenin (Plakoglobin), ENO1, ALDOA and GAPDH was confirmed by RNA immunoprecipitation (RIP). ENO1, enolase; ALDOA, aldolase; GAPDH, glyceraldehyde 3-phophatedeshydrogenase. The results are the median of three technical replicates; error bars represent S.D. (**E**) Subcellular localization of *lnc-Nr6a1* isoforms. Expression of *Hprt*, *lnc-Nr6a1-1*, *lnc-Nr6a1-2*, *U3* and *Xist* in cytoplasmic, nuclear soluble and nuclear insoluble fractions of NMuMG cells. The graph is an average of three technical replicates; error bars represent S.D.

**Figure 8 ncrna-08-00062-f008:**
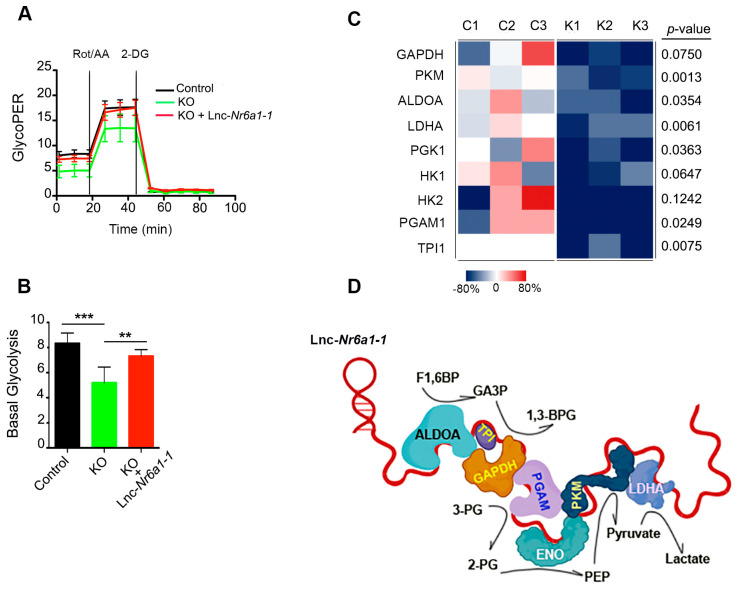
Defective glycolysis in *Lnc-Nr6a1*-depleted cells. (**A**) Proton Efflux Rate derived from glycolysis (glycoPER) of the control, *Lnc-Nr6a1*-depleted cells and *Lnc-Nr6a1*-depleted cells overexpressing the *lnc-Nr6a1-1* isoform. Rot/AA Rotenone/antimycin, 2-DG 2-Deoxy-D-glucose. (**B**) Plots of glycoPER quantification corrected using total µg of protein. GlycoPER is expressed as pmol/min/µg protein. Data from five independent experiments are presented as mean ± S.D. *** *p* < 0.001, ** *p* < 0.01 by two-tailed Student’s *t*-test. (**C**) Enrichment of glycolytic enzymes after immunoprecipitation with enolase-1. The percent change with respect to the average spectral count for the corresponding control group is indicated. A two-group, two-tailed Student’s *t*-test was used to determine the statistical significance of differences between the control (C_1–3_) and *Lnc-Nr6a1*-depleted cell (K_1–3_) samples. GAPDH, glyceraldehyde 3-phophatedeshydrogenase; PKM, pyruvate kinase; ALDOA, aldolase; LDHA, lactate dehydrogenase; PGK, phosphoglycerate kinase; HK, hexokinase; PGAM, phosphoglycerate mutase; TPI, triosephosphate isomerase (**D**) A proposed model for the role of *lnc-Nr6a1-1* as a scaffold molecule involved in the formation of a glycolytic complex. F1,6BP, fructose-1,6-biphosphate; GA3P, 3-phosphoglycerate; 1,3BPG, 1,3-bisphosphoglycerate; 3-PG, 3-phosphoglycerate; 2-PG, 2-phosphoglycerate; PEP, phosphoenolpyruvate.

## Data Availability

NGS data have been deposited to Gene Expression Omnibus (GEO) under the accession number GSE207917. GSE207915 and GSE207916 correspond to accession numbers of ChIP-seq data and RNA-seq data, respectively. (These data can be accessed through the link https://www.ncbi.nlm.nih.gov/geo/query/acc.cgi?acc=GSE207917 (accessed on 4 August 2022) with the Reviewers Token: ojkxyiayxbudbsz).

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
