# Peer review of "TGF-β-Upregulated *Lnc-Nr6a1* Acts as a Reservoir of miR-181 and Mediates Assembly of a Glycolytic Complex"

_ncrna, 2022, doi:10.3390/ncrna8050062_

Round 1

Reviewer 1 Report

The manuscript "TGF-β-upregulated Lnc-Nr6a1 acts as a reservoir of miR-181 and mediates assembly of a glycolytic complex" describes a comprehensive study of the functions of long non-coding RNA Lnc-Nr6a1 in mice. The study was carried out at the highest level using modern methods, described in detail by the authors, the manuscript is accompanied by informative figures and tables, including supplementary materials confirming the objectivity of the study and the conclusions of the authors.

However, I have some minor comments:

1. The word "embebed" is repeated twice in the manuscript, in lines 81 and 299. I would ask the authors to check the spelling of this word. Probably there is a typo.

2. In the chapter "Materials and Methods" the authors mention TRIzol Reagent (Invitrogen) and TRIzol (Thermofisher Scientific). Did the authors use reagents from different companies? If yes, please explain why.

It is also desirable to bring the names of reagent manufacturers to uniformity, in some cases the authors write only the name of the company, and in some cases the city of reagent manufacturers as well.

3. See line 679. The text says "a 4°C", probably it is a typo.

4. Please, correct line 724, there is a typo.

5. I would ask the authors to carefully reread the chapter "Materials and Methods", since there are quite a few typos in it, for example, there are repeated spaces, and this is striking.

Reviewer 2 Report

Great study conducted by the authors to delineate the biological function of lncRNA, Lnc-Nr6a1 during EMT. It still remains unclear in what disease conditions this study might hold high value. In other words, is the study only focused on the molecular characterization of Lnc-Nr6a1?

Why did authors specifically chose  Nr6a1os from the microarray dataset?

What is the reasoning for detecting lnc-Nr6a1-1 isoform  in both RNA (poly A- and poly A+) populations? Was there any poly A pulldown assay performed on this RNA fractions

How did the authors decide on anoikis? Was there any functional and phenotypic screening conducted in epithelial cells?

All the figures need extensive, careful and major revision. The font size and figure sizes are very small. There is plenty of space left in each page for the figures to improve their legibility. There is no statistical analysis and presentation of significance values. The color schemes are confusing and would urge the authors to modify them to better color schemes, with darker annotations. Figures have no labels! For instance, figure 3A- what gene are we looking at? 3B- What amplicon is presented? what are the bands and their sizes, marker is missing? 3E (volcano plot)- highlight some of the key genes in the down and upregulated spots. 3F- what do the scientific format labels define for each of the GO terms? 

In all the western blots, it is surprising to see that the beta-actin signals are low or faint. Why is this the case when there are several well characterized antibodies for this protein? Please repeat and present better blots.

Another confusion in the figures is the method of presenting the gene names. For instance, if its a mRNA it must be italicized. That way one would know if its mRNA or not. If its protein the whole name must be in UPPERCASE.

Figure 2, no relative intensity analysis is presented for the protein targets analyzed by western blotting! Same applies to other figures

There is a lack of negative and positive controls in most of the experiments that are reported in the manuscript! 

Imaging results need quantification from multiple trials clearly detailed in legends. 

The figure legends must be revised for each figure with relevant and correct statistics. So, as the p-value annotations in the figures. There is no mentioning of the number of experimental trials or replicates.

The discussion section needs a major revision after revising the figures and results. The manuscript needs careful and thorough attention and all the comments must be addressed. 

Reviewer 3 Report

In this manuscript, the authors studied the biogenesis of a conserved (from mouse to human) LncRNA and its role in cell adhesion. Combining multiple assays, including RNA-seq, interactome, biochemistry, and genetic approaches, they found that lnc-Nr6a1-1 and lnc-Nr6a1-2 are upregulated after TGF-beta treatment through different signaling pathways. Furthermore, knocking out Lnc-Nr6a1 leads to a significant gene expression change, mainly affecting the cell adhesion and glycolytic process. Consistently, they found that Lnc-Nr6a1 depletion affects cell adhesion in both at long and short time, and activation of Lnc-Nr6a1 expression will affect the protein level of E/N-cadherin. Lastly, by using iDRIP-MS, they found that Lnc-Nr6a1 may serve as a scaffold for the formation of the glycolytic complex. Overall, this is a comprehensive study and explored many different perspectives regarding the biogenesis and physiological function of Lnc-Nr6a1, although the clear mechanism is still missing (see comments below). For improvements of this manuscript, I have the follow suggestions/comments:

Major points:

1. The mechanism how Lnc-Nr6a1 regulates cell adhesion is not fully explored. Although the authors showed that in Lnc-Nr6a1 depletion cells, the genes in the cell adhesion pathway are downregulated, leading to reduced cell adhesion, what is the function of Lnc-Nr6a1 in this pathway is not well studied. To at least fully confirm its role in cell adhesion, the authors should perform some experiments to characterize the genetic relationship between Lnc-Nr6a1 and certain genes (which could be the genes identified in RNA-seq) involved in cell adhesion.

2. Does Lnc-Nr6a1 depletion affect the expression of its host gene NR6A1? If so, is it possible that at least some the differently expressed genes identified could be due to this, since NR6A1 has been reported as a transcription factor (PMID: 31315616) and has been implied in E-cadherin regulation(PMID: 33034957). Although the authors showed convincing data that the cell adhesion defect caused by Lnc-Nr6a1 depletion can be rescued by the overexpression of Lnc-Nr6a1, it is still important to rule out/investigate whether this phenotype could be coordinated by both sense and anti-sense genes.

3. Fig.8C. The protein level of each component need to be checked since the assay (IP+mass spec) used in this panel does not rule out the possibility that the reason less protein complex gets enriched after Lnc-Nr6a1 depletion is simply because it leads to lower protein level of some components, instead of affecting the complex formation.

4. Fig.3D. Since miR181a2 and b2 are processed from the same hosting mRNA, why does knocking out Lnc-Nr6a1 cause the different responses in the biogenesis of these two miRNAs? (increased a2-3p but reduced b2-3p). Also, in the last panel of 3D, the y-axis is not labelled clearly. Is it a1 or a2?

5. Fig.6D. Why the effect on E/N-cadherin are different between Lnc-Nr6a1-1 OE and Guide2/8, since they both essentially increased the level of Lnc-Nr6a1-1. Does it mean this effect is from the Lnc-Nr6a1-2 rather than 1-1? Also, the N-cadherin blot pattern does not match the quantification: it does not appear to be 1.5 fold change compared to the control. The authors need to include more repeats and statistics to back up this conclusion.

Minor points:

1. The authors need to add more instruction on background information. For example, if I understand correctly, EMT stands for epithelial-mesenchymal transition. However, there is nowhere in the text to mention this, and its physiological relevance. The authors should not assume all the readers to be experts in this area.

2. Line 160-161. CHX chase is a good experiment to rule out the effect from secondary cascade, but still not the direct evidence to confirm they are the direct targets. So here should only be a suggestion.  

3. Fig.2E, the y-axis of the last panel, should be S1P1.

Round 2

Reviewer 2 Report

Dear Authors,

The revised manuscript looks far better than the original. Few major points-

1) It seems like the font sizes in the figures are not revised in the new version, as they look the same with original.

2) Second, please revise the figure 7! what is the difference between the old and the new versions? Especially modify 7D color scheme. Does this figure demonstrate the pulldown efficiency between IgG and lnc-Nr6a1-1? If so, then kindly use a binary color scheme. For instance, for IgG we there is a turquoise label for all targets similarly keep it one color for lnc-Nr6a1-1.Otherwise it is very confusing. 

3) For figure 8 D, provide the model in a bigger size, as it is the summary of the entire findings of the study. May be use it as a graphical abstract with a few key highlights summarizing the main findings and its importance. 

4) It is important to provide the densitometry analysis for figure 2 western blots. Including the analysis will not be cumbersome rather entail the overall direct and indirect effects of inhibitors. It is important to ensure the robustness of these findings.  There is still lot of empty space in the page for figure 2 that can be very well utilized. Please plan accordingly. A simple solution is that the figure panels can be designed in horizontal instead of vertical. 

Reviewer 3 Report

In this revised manuscript, the author addressed most of the concerns raised by this reviewer by performing additional experiments and editing the wording. Overall, the data presented in the manuscript now is largely supporting the statements and should be suitable for publication. 

Author Response

Thank you for your insightful and constructive comments.